# Overexpression of *EgrIAA20* from *Eucalyptus grandis,* a Non-Canonical *Aux*/*IAA* Gene, Specifically Decouples Lignification of the Different Cell-Types in *Arabidopsis* Secondary Xylem

**DOI:** 10.3390/ijms23095068

**Published:** 2022-05-03

**Authors:** Hong Yu, Mingjun Liu, Zhangsheng Zhu, Aiming Wu, Fabien Mounet, Edouard Pesquet, Jacqueline Grima-Pettenati, Hua Cassan-Wang

**Affiliations:** 1Laboratoire de Recherche en Sciences Végétales, Université de Toulouse, CNRS, UPS, Toulouse INP, 31320 Auzeville-Tolosane, France; hongyu@swmu.edu.cn (H.Y.); mingjunliu@pitt.edu (M.L.); zhuzs@scau.edu.cn (Z.Z.); fabien.mounet@univ-tlse3.fr (F.M.); grima@lrsv.ups-tlse.fr (J.G.-P.); 2State Key Laboratory for Conservation and Utilization of Subtropical Agro-Bioresources, Guangdong Key Laboratory for Innovative Development and Utilization of Forest Plant Germplasm, College of Forestry and Landscape Architectures, South China Agricultural University, Guangzhou 510642, China; wuaimin@scau.edu.cn; 3Arrhenius Laboratories, Department of Ecology, Environment and Plant Sciences (DEEP), Stockholm University, Svante Arrhenius väg 20A, 106 91 Stockholm, Sweden; edouard.pesquet@su.se

**Keywords:** secondary xylem, auxin, non-canonical *Aux/IAA*, secondary fiber, cambium differentiation, *Eucalyptus*, syringyl lignin, IAA20, wood, *Arabidopsis*

## Abstract

Wood (secondary xylem) formation is regulated by auxin, which plays a pivotal role as an integrator of developmental and environmental cues. However, our current knowledge of auxin-signaling during wood formation is incomplete. Our previous genome-wide analysis of *Aux*/*IAAs* in *Eucalyptus grandis* showed the presence of the non-canonical paralog member *EgrIAA20* that is preferentially expressed in cambium. We analyzed its cellular localization using a GFP fusion protein and its transcriptional activity using transactivation assays, and demonstrated its nuclear localization and strong auxin response repressor activity. In addition, we functionally tested the role of *EgrIAA20* by constitutive overexpression in *Arabidopsis* to investigate for phenotypic changes in secondary xylem formation. Transgenic *Arabidopsis* plants overexpressing *EgrIAA20* were smaller and displayed impaired development of secondary fibers, but not of other wood cell types. The inhibition in fiber development specifically affected their cell wall lignification. We performed yeast-two-hybrid assays to identify *EgrIAA20* protein partners during wood formation in *Eucalyptus*, and identified EgrIAA9A, whose ortholog PtoIAA9 in poplar is also known to be involved in wood formation. Altogether, we showed that *EgrIAA20* is an important auxin signaling component specifically involved in controlling the lignification of wood fibers.

## 1. Introduction

Wood, or secondary xylem, is one of the most abundant natural and renewable sources of energy, as well as an important bio-resource for global industry. It provides raw material for construction and building, pulp/paper making, energy production as well as a source of fine chemicals ranging from plasticizers to flavoring agents. Wood is one of the most environmental-friendly resources that enables both to replace our dependency on fossil fuels and to capture atmospheric CO_2_ associated with global warming actively [1]. In trees, wood complex organization functions as a vascular/skeletal tissue due to the presence of several specific cell types characterized by the presence of secondary cell walls (SCW). These include the sap-conducting cells called tracheary elements, the skeletal support cells called fibers, and ray cells allowing to load and unload the sap content. These wood cell types all derive from the differentiation of a lateral secondary meristem called the vascular cambium that undergoes orientated cell division, cell expansion, and massive deposition of SCW polysaccharides in patterns guided by microtubules, programmed cell death preceding or not cell wall lignification depending on the xylem cell type [2]. Albeit following the same general blueprint, each xylem cell type has specific cytological features (i.e., pattern and composition of SCWs) to fulfill their roles such as in the case of lignin. This cell wall polymer is composed of phenylpropanoids that vary in both their terminal aliphatic functions (alcohol, aldehyde, acid) and their level of aromatic substitutions (meta hydroxy/methoxylation). Canonical lignin residues include *p*-hydroxyphenyl (H) with no meta substitution, guaiacyl (G) with one meta substitution, and syringyl (S) with two meta substitutions [3]. Each distinct xylem cell type and the different cell wall layers are enriched with specific lignin residues. The SCWs in tracheary elements are enriched in G residues; the fiber SCWs are enriched in S residues, whereas the primary cell wall separating these cell types are enriched in H residue [4,5]. The succession of the differentiation stage of xylem cell types from their initiations and their coordination in time of specific cytological features to their full maturation depend on growth factors that have either overlapping or distinct effects and that depend on the xylem cell types, their differentiation stage, and/or the surrounding environmental constraints [6,7,8]. Among these growth factors, an increase in auxin levels plays an essential role for tracheary element initiation [9,10], but also uncouples xylem cell types—not affecting TE formation but fibers to give rise to either in parenchyma cells or fibers with a reduced SCW formation [11,12]. Yet, the molecular mechanisms allowing auxin to uncouple the formation of the different xylem cell types remain unclear.

Auxin signaling is thus capable of directing cellular behaviors from cell division, expansion, and differentiation [13,14]. However, our current knowledge of the auxin signaling pathway components involved in secondary xylem differentiation remains incomplete. Auxin signaling is mediated by two classes of transcription factors: AUXIN/INDOLE-3-ACETIC ACID (*Aux*/*IAA*) and AUXIN RESPONSE FACTOR (ARF) [15]. In the absence of auxin, the *Aux*/*IAA* repressors bind to the ARFs to stop transcription. In contrast, when auxin is present, *Aux*/*IAA*s associate with S-PHASE KINASE-ASSOCIATED PROTEIN1 SKP-Cullin-F BOX (SCF) ^TRANSPORT INHIBITOR RESISTANT1/AUXIN SIGNALING F-BOX^ (SCF^TIR1/AFB^), which triggers their ubiquitination and proteolysis. The interacting ARF transcription factors are thus freed and able to modulate the expression of the downstream auxin-responsive genes [16,17]. Despite numerous studies showing the role of ARF5/MONOPTEROS (ARF5/MP) and its interacting *Aux*/*IAA* members during primary xylem formation (i.e., IAA12 in embryogenesis, IAA20/30 in root and leaf) [18,19,20,21,22], only a handful of studies have investigated the role of *Aux*/*IAA* members and their interacting ARFs during secondary xylem formation [23]. These studies include the role of *PttIAA3* in hybrid aspen, which is highly expressed in wood forming tissue. The overexpression of a stabilized *PttIAA3* version reduced the width and the length for both tracheary elements and fibers in transgenic lines in addition to a reduction in cambial cell division [24]. *EgrIAA13* in *Eucalyptus* and its orthologs in *Populus* are preferentially expressed in the xylogenic tissues and downregulated in tension wood. The downregulation of *EgrIAA13* using the RNAi knockdown construct impeded *Eucalyptus* xylem fiber and vessel development in Induced Somatic Sector Analysis experiments [25]. The *Eucalyptus EgrIAA4*, which is highly expressed in cambium, was shown to uncouple xylem cell types, maintaining the formation of tracheary elements but altering fibers when overexpressed in *Arabidopsis* [26]. It is worth noting that *Arabidopsis* undergoes true secondary growth wood formation (i.e., originating from cambial activity), although devoid of ray cells, in hypocotyls, roots, and at the basal part of inflorescence stems when stimulated by short-day growth conditions [27,28]. The formed secondary xylem in hypocotyls comprises two distinct phases: (i) phase I with xylem composed mainly of tracheary elements and parenchyma cells, and (ii) phase II with both tracheary elements and fibers [28]. *Arabidopsis* can thus be used as a reliable simplified model of wood formation to investigate the function of key genes in trees, from *Eucalyptus* to poplar, using heterologous over-expression [26,29,30].

Within the genome of *Eucalyptus grandis* [31], several candidate *ARF* and *Aux*/*IAA* genes are associated with wood formation. Among them, *EgrIAA20*, a non-canonical *Aux*/*IAA* member, is preferentially expressed in cambium and developing secondary xylem of *Eucalyptus*. Herein, we performed the functional analysis of the atypical *EgrIAA20* that lacks the highly conserved domain II required for the rapid degradation of *Aux*/*IAA* protein in the presence of auxin. Using transient expression in tobacco protoplast, we showed that *EgrIAA20* functions as a strong transcriptional repressor of auxin responses. The overexpression of *EgrIAA20* not only impaired vascular patterning in cotyledons during primary growth, but also altered secondary xylem development by altering secondary fiber development but not tracheary elements. Screening for the protein partners of *EgrIAA20* during wood formation identified EgrIAA9A as the main interacting protein. This suggests that a complex EgrIAA9A-*EgrIAA20* is likely to be involved in specific wood cell type differentiation. Our study reveals that the ubiquitous and substantial over-expression of the auxin-dependent transcription factor *EgrIAA20* specifically uncoupled the differentiation and/or lignification of tracheary elements from fibers. This novel discovery not only opens new avenues to deepen our understanding about the regulation of wood cell type differentiation, but also provides a proof-of-concept strategy to control wood cell types and thus lignin composition without affecting sap conduction.

## 2. Results

### 2.1. EgrIAA20 Transcripts Accumulate Preferentially in Cambium in Eucalyptus

We previously identified 26 *Aux/IAA* genes in the *E. grandis* genome and numbered them according to their phylogenetic relationship with their *Arabidopsis* orthologs [26]. Large scale RT-qPCR expression analyses of these genes across various tissues and organs, including cambium and developing xylem, enabled us to identify 11 members with significantly high expression levels in vascular tissues [26]. Among these *Aux*/*IAA* paralogs, *EgrIAA20* presented the highest expression level in vascular tissues, especially in cambium and xylem (Figure 1A,B), followed by *EgrIAA19*, *EgrIAA31*, *EgrIAA4*, and *EgrIAA13*. Complementary analyses using the EucGenIE database, an online resource for *Eucalyptus* genomics and RNAseq transcriptomics, further confirmed the preferential expression of *EgrIAA20* in tree trunks (Figure 1C). We therefore selected *EgrIAA20* as a promising candidate likely involved in the auxin regulatory mechanisms controlling the differentiation of wood cell types.

### 2.2. EgrIAA20 Defines a Non-Canonical Aux/IAA Protein Which Lacks Highly Conserved Aux/IAA Characteristic DOMAIN II

Protein sequence analyses and alignments between *Aux*/*IAA* para/orthologs revealed that *EgrIAA20* lacks the highly conserved Domain II (Figure 2A,C). Domain II is conserved in the majority of the *Aux*/*IAA* para/orthologs, and single mutations in this domain increase protein stability thus hampering normal auxin responses that rely on the rapid degradation of *Aux*/*IAA* [32]. The closest *Arabidopsis* orthologs of *EgrIAA20* are AtIAA20 and AtIAA30, which also lack Domain II. These non-canonical *Aux*/*IAA* proteins such as AtIAA20 were shown to be long-lived protein in *Arabidopsis* in contrast to the other paralogs [33]. It is worth noting that non-canonical *Aux*/*IAA* devoid of Domain II are not present in grapevine [34], nor in poplar species such as black cottonwood (PtrIAA20.1, PtrIAA20.2) or aspen (PttIAA3) [24] (Figure 2A–C) in this phylogenetic clade. Additional analyses also revealed the presence of the highly conserved Domain I with the repression motif TDLRLGL. The core motif “LxLxL” is also called EAR repression motif (Ethylene Response Factor (ERF)-Associated amphiphilic Repression), which can recruit a TOPLESS (TPL) transcriptional co-repressor [35]. We thus identified unique *Aux*/*IAA* proteins, lacking Domain II controlling their turnover rate and with a potential EAR repression motif, present in both *Arabidopsis* and *Eucalyptus*.

### 2.3. EgrIAA20 Is Mainly but Not Exclusively Nuclear Localized

In general, *Aux*/*IAA* proteins have two types of nuclear localization signals (NLSs). The type I NLS is a bipartite structure comprising a conserved KR basic doublet located between Domain I and II and associated with the basic amino acids from Domain II. The type II NLS corresponds to a basic residue-rich region located in Domain IV that resembles the SV40-type NLS (Figure 2A). Protein sequence analysis showed that *EgrIAA20* contains a type II NLS in Domain IV but lacks type I NLS (Figure 2A). We therefore investigated the subcellular localization of *EgrIAA20* using transient expression in tobacco protoplasts. To ensure the exclusive nuclear targeting, we used the canonical *Aux*/*IAA* EgrIAA4 that harbors two conserved NLSs as a reference [26]. As expected, fluorescence microscopy analysis confirmed that EgrIAA4 fused to GFP was exclusively targeted to nuclei (Figure 3A). In cells transformed with *GFP* alone, a fluorescence signal was found throughout the cell, in both nuclei and cytoplasm, thus confirming the functionality of NLS in EgrIAA4 (Figure 3A). For *EgrIAA20* fused to *GFP*, the fluorescence signal was found mostly in nuclei but also in the cytoplasm with a less intense signal than with the GFP alone (Figure 3A). This extended localization of *EgrIAA20* in the extra-nuclear compartment is probably due to the lack of bipartite NLS, as similar wide localization profiles were also reported for tomato non-canonical *Aux*/*IAA* member SlIAA32 also lacking the type I NLS [36]. To our knowledge, the function ensured by IAA20 and its orthologs in the cytosol remains unknown; further experiments are needed to explore its function based on this cytosolic localization. We thus confirmed that *EgrIAA20* is mostly localized in nuclei to mediate auxin response similar to other *Aux*/*IAA* paralogs.

### 2.4. EgrIAA20 Acts as a Potent Transcriptional Repressor of Auxin Response In Vivo

To ensure that *EgrIAA20* effectively affected auxin responses, we evaluated its transactivation activity using transient expression experiments in tobacco BY-2 protoplasts co-transfected with a *DR5* auxin responsive-promoter controlling *GFP* as a reporter gene (Figure 3B). This *DR5::GFP* construct is a synthetic auxin responsive-promoter composed of several copies of the TGTCTC core motif of auxin response *cis* element [37]. The GFP activity of the *DR5::GFP* reporter was highly induced by auxin treatment (50 µM 2,4-D) and not affected by the co-transfection with a mock effector (empty plasmid containing the *CaMV 35S* promoter without the effector gene) (Figure 3B). Auxin treatment combined with co-transfecting the canonical *Aux*/*IAA* protein EgrIAA4 showed a strong repression of GFP fluorescence intensity, as previously demonstrated [26]. Co-transfection with *EgrIAA20* together with auxin treatment also reduced GFP fluorescence (12% of residual activity when compared to the mock effector) (Figure 3B). This result revealed that *EgrIAA20* harboring an EAR repression motif was effectively able to mediate auxin response in vivo and acted as a strong transcriptional repressor.

### 2.5. Over-Expression of EgrIAA20 in Transgenic Arabidopsis Results in Smaller and Bushier Plants with Floppy Stems

We then investigated the functional role of *EgrIAA20* in secondary xylem differentiation in planta by constitutively and ubiquitously overexpressing *EgrIAA20* in *Arabidopsis*. This approach is similar to previously successful functional analyses of *Eucalyptus* genes involved in the regulation of wood formation [26,30]. In general, the overexpression of the wild-type *Aux/IAA* genes results in no significant phenotypic changes, which were explained by the intrinsic rapid turnover [38]. The effect of *CaMV 35S* promoter-driven *EgrIAA20* was compared to plants transformed with empty vectors. More than 10 independent lines were obtained, all exhibiting reduced plant size compared to mock transformed plants. We chose three phenotypically representative transgenic lines to assess *EgrIAA20* mRNA accumulation. A higher expression level of *EgrIAA20* was detected in all the transgenic lines, with line 3.1 exhibiting the highest expression level compared to lines 1.3 and 3.3 (Figure 4A). We thus selected the strongest (Line 3.1) and the weakest (Line 3.3) expressing lines to characterize the effects of ubiquitous overexpression of *EgrIAA20* in *Arabidopsis*.

Strong phenotypic alterations were triggered by *EgrIAA20* overexpression (Figure 4B), some dependent and other independent from transgene expression levels, each affecting differently specific organs. Leaves and rosette diameter were significantly reduced in transgenic plants compared to mock controls (Figure 4C,D). Rosette leaves also displayed helical twisting and backward rolling in early stages of development (Figure 4B). Using vertically grown seedlings in vitro, root growth was also impacted with *EgrIAA20* overexpressing plants showing significant reduction in primary roots’ elongation as well as a limited formation of lateral roots compared to controls (Figure 4E,F). For stems, although bolting time was not affected, their growth rates were significantly reduced in overexpressing plants compared to control plants, especially in the strong line 3.1 (Figure 5A–C). As a consequence of reduced inflorescence stem growth, the *EgrIAA20* overexpressing lines displayed a shorter and bushier morphology (Figure 5A,B), accentuated accordingly with the overexpression levels to 28% and 81% reduction in lines 3.3 and 3.1, respectively (Figure 5C). The inflorescence stems of the transgenic plants were also thinner than the WT. The diameter of the basal part of the inflorescence was significantly reduced in transgenic lines (46% in line 3.1 and 29% in line 3.3; Figure 5D), although the ratio of stem height to width did not significantly differ between genotypes. Remarkably, the *EgrIAA20* overexpressing lines displayed loss of stem rigidity in contrast to the upright stems of controls (Figure 4A,B), suggesting a likely alteration of skeletal support similarly to mutants such as *ifl1* with impaired fiber formation in secondary xylem [39]. Last, the flowering time in *EgrIAA20* overexpressing lines was delayed. The flowers from transgenic plants remained as closed buds for longer (>10 days) and generally failed to reach the anthesis stage. Closer observations showed that the stamens were dramatically shorter than in the control plants (Figure 5E). The fertility of *EgrIAA20* overexpressing lines was also severely decreased often leading to barren fruits (0 to 20 seeds per transgenic plants compared to more than 200 seeds per WT plants). However, pollination of the transgenic lines using the WT pollen enabled seed formation, indicating that the ovules of *EgrIAA20* overexpressing lines were fertile. Altogether, *EgrIAA20* overexpressing plants showed phenotypic defects similar to *ref3*/*c4h* mutants that are affected in the lignin biosynthetic *CINNAMATE-4-HYDROXYLASE* gene with reduced stem size, loss of stem rigidity, reduced stamen filament growth, and barren fruits [40]. Thus, our results suggest that secondary xylem cell type and/or their lignin formation could be altered by the overexpression of *EgrIAA20.*

### 2.6. EgrIAA20 Over-Expression Reduces Fiber-Specific Lignin in Arabidopsis Hypocotyls

To investigate the causes of the reduced growth and loss of rigidity due to *EgrIAA20* overexpression, we analyzed the chemical composition of cell walls in ball-milled hypocotyls with pyrolysis gas chromatography/mass spectrometry (Py-GC/MS). The *EgrIAA20* overexpressing lines showed no changes in cell wall polysaccharides compared to WT plants (Table 1). In contrast, a significant reduction in lignin S residue content as well as of other phenolic residues was observed, together with an increase in lignin H residues but no changes in lignin G residue content compared to WT plants (Table 1). Consequently, changes in lignin composition shown by the S/G ratio were largely reduced by the overexpression of *EgrIAA20* (Table 1). Altogether, our results show that the constitutive and ubiquitous overexpression of *EgrIAA20* altered specifically the accumulation of S residues in lignin and not the other cell wall components.

### 2.7. EgrIAA20 Overexpression Alters Primary Xylem Development in Cotyledons

To assess the role of *EgrIAA20* overexpression on primary xylem development, we investigate the vascular patterning of cotyledons by measuring the number of secondary vein loops from the mid-vein as previously reported [41]. In control seedlings, cotyledons’ vein patterns belonged to classes I and II, and 45% of them exhibited a more complex pattern (Class I with four complete loops, Class II with at least two complete loops) (Figure 6A). In contrast, the cotyledons’ vein patterns of all *EgrIAA20* overexpressing lines exclusively belonged to classes II and III (class III showing no complete vascular loop), and none of them belonged to class I. This result indicated that the overexpression of *EgrIAA20* prevented the completion of vascular patterning during primary growth.

This incomplete vascular patterning in cotyledon phenotype mimics that of *Arabidopsis ARF5* loss-of-function mutant *monopteros/arf5.* Considering that *AtARF5* was reported to control this development process in *Arabidopsis* [19,42,43], we tested if *EgrIAA20* interacts with AtARF5 and its potential *Eucalyptus* ortholog EgrARF5, using yeast two-hybrid. The results confirmed that there was a strong interaction between *EgrIAA20* and AtARF5, as well as between *EgrIAA20* and EgrARF5 (Figure 6B), which further supported the orthologous relationship between *Arabidopsis* and *Eucalyptus* defined by our phylogenetic analysis [26,44]. We thus also confirmed that the interaction between *Aux*/*IAA* and ARF is conserved between *Arabidopsis* and *Eucalyptus*.

### 2.8. EgrIAA20 Overexpression Specifically Represses Xylary Fibers of Secondary Xylem in Arabidopsis Stem and Hypocotyl

The secondary growth producing secondary xylem cells occurs naturally in adult plants in the base part of the inflorescence stem, root and hypocotyl [27]. To investigate if the overexpression of *EgrIAA20* also impacted secondary xylem cells’ development, we particularly examined the basal part of inflorescence stems (<1 cm to the rosette level) as well as hypocotyls of adult *Arabidopsis* plants using a short-day condition to stimulate secondary growth [28]. To detect primary and secondary xylem (all containing lignified SCW) in cross-sections easily, we used phloroglucinol/HCl or the Wiesner test that specifically stains the lignified SCW into red-purple by reacting with lignin coniferaldehyde residues [45], or the Mäule test that allows the staining of the SCW of the tracheary element in brown and fiber in red [46]. Cell types present in the primary xylem cells include tracheary elements, xylary fibers, and interfascicular fibers (Fiber I) [47], whereas secondary xylem cells mainly include secondary tracheary elements (Vessel II) and secondary fibers (Fiber II) (Figure 7C and 7D). In WT plants, cell walls of all primary and secondary xylem cells were all stained positively by the Wiesner test (Figure 7A), with some tracheary elements exhibiting more intense staining (Figure 7C, indicated by the blue arrow). In contrast, the intensity of the Wiesner staining was dramatically reduced in *EgrIAA20* overexpressing lines (Figure 7E–L), indicating a modified lignification with a reduced coniferaldehyde incorporation [45]. Additionally, the strongest overexpressing lines 1.3 and 3.1 also showed a complete absence of staining in secondary fibers but not in the primary fibers (Figure 7E–L). Only a few isolated stained cells were observed in the secondary growth region (both interfascicular and vascular bundle regions) (Figure 7E–L). In order to confirm the cell type identity, complementary staining was performed using the Maüle test to discriminate fiber cells stained in red due to SCW enriched lignin S residue, from tracheary element cells stained in brown due to SCW enriched in lignin G residue (Figure 7D). The isolated cells in the secondary region of transgenic plants were all stained in brown, confirming their tracheary element identity (Figure 7H,L). Moreover, cells’ in-between tracheary elements in the secondary xylem zone of *EgrIAA20* overexpressing lines (Figure 7H) did not exhibit the red staining observed in WT plants (Figure D), thus confirming that S residue incorporation was affected in transgenic plants. The moderate level of overexpression of line 3.3 exhibited an intermediate phenotype between WT and strongly overexpressing lines (Figure 7M–P).

Further analyses on adult plant hypocotyls were performed due to its strong similarity with wood in perennial species such as poplar [28]. In WT plants, we confirmed the normal biphasic development of secondary xylem in short-day growth conditions with an early phase (Phase I), in which lignified tracheary elements are formed but not fibers (Figure 8A,B), followed by a late phase (Phase II), in which both lignified fibers and tracheary elements are formed [28]. However, secondary xylem lignification was greatly altered in *EgrIAA20* overexpressing lines: the Wiesner test staining showed normal staining for tracheary elements in Phase I and II in WT, but an absence of staining of fibers in Phase II of transgenic lines, independently of the *EgrIAA20*-overexpression level (Figure 8E–P). Complementary staining using the Maüle test confirmed the presence of tracheary elements enriched in G lignin, but also the complete absence of S lignin enriched secondary fibers in *EgrIAA20* overexpressing lines (Figure 8H,L,P). Overall, our results revealed that the lignification of secondary fibers is specifically reduced by the dose-dependent overexpression of *EgrIAA20* as well as uncoupled from the lignification of tracheary elements.

**Figure d64e1088:**
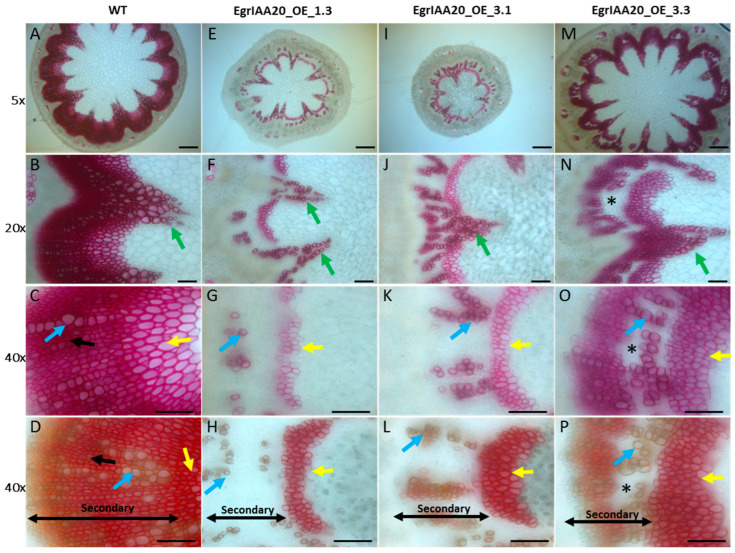


### 2.9. Identification of EgrIAA9A as the Main Interacting Protein of EgrIAA20

To deeper our understanding of the molecular modality of the *EgrIAA20* regulation mechanism during wood formation in trees, we aimed to identify its interacting protein partners. To identify proteins that interact with *EgrIAA20* during wood formation in *Eucalyptus*, we screened a *Eucalyptus* wood yeast two hybrid (Y2H) library [30] with *EgrIAA20* as a bait. We successfully sequenced 35 clones among the 47 positive clones of Y2H (Appendix A), showing that the most likely interacting candidate was EgrIAA9A, which represented 10 out of 35 total sequenced colonies (29%). This interaction was confirmed by targeted Y2H between *EgrIAA20* and EgrIAA9A, independently of *EgrIAA20* being used as bait or prey (Figure 9). Three other *Aux*/*IAA* genes were also identified: IAA1, IAA11, and IAA16 (two clones) with lower representation. Two out of 35 clones corresponded to a *Eucalyptus* ortholog of *SGT1* (*SUPPRESSOR OF THE G2 ALLELE OF* *SKP1*), previously reported to be a positive regulator of auxin signaling and disease resistance in *Arabidopsis* [48]. Strikingly, no ARF proteins were identified from our Y2H screening although they are well-known interacting proteins of *Aux*/*IAA* proteins and were identified above as interacting with *EgrIAA20* for EgrARF5 (Figure 6B). We therefore performed a targeted Y2H to determine any interaction between *Eucalyptus* wood-expressing-ARF proteins [44] and *EgrIAA20*. The results showed that *EgrIAA20* interacted with all four wood-related EgrARFs: EgrARF4, EgrARF5, EgrARF10, and EgrARF19 (Figure 9). Our targeted Y2H results also showed that *EgrIAA20* was able to form homodimers with *EgrIAA20*, and confirmed the strong interaction with EgrIAA9A independently of its position as bait or prey (Figure 9). Overall, our results suggest that *EgrIAA20* may form a transcription regulator complex preferentially with EgrIAA9A as well as other *Aux*/*IAA* but also with ARFs.

## 3. Discussion

In this study, we performed the functional characterization of a novel non-canonical *Aux/IAA* member, *EgrIAA20*, which is preferentially expressed in wood cambium. This non-canonical *Aux/IAA* was characterized by the absence of Domain II, pointing to a lower turnover compared to the canonical short-lived paralogs, as well as an EAR transcriptional-repression motif on its Domain I (Figure 2). We experimentally confirmed these predictions by showing that *EgrIAA20* was mostly targeted to nuclei and presented a repressor activity on auxin-responsive transcription (Figure 3). As genetic engineering is strongly hampered in eucalypts species, we performed gain-of-function studies of *Eucalyptus* genes directly in *Arabidopsis* as previously achieved [26,30]. We ensured the biological relevance of the heterologous *EgrIAA20* overexpression in *Arabidopsis* by confirming the conservation of both its sequence structure, both lacking Domain II and presenting the EAR motif (Figure 2), and its interactions with *ARF5* independently of the species used (Figure 6B). This result thus confirmed that true orthologs exist between *Arabidopsis* and *Eucalyptus*, unlike other angiosperms which have lost these specific paralogs (Figure 3). Homologous gain-of-function experiments in *Arabidopsis* plants by overexpressing *AtIAA20* [49], the ortholog of *EgrIAA20* [26], showed defects in vascular patterning in cotyledon [49] similar to the severe defects observed in *EgrIAA20* overexpressors (Figure 6). However, previous studies did not evaluate the impact of the genetic modulation of *AtIAA20* on secondary xylem formation or lignification as we performed in the herein study. Although the sweeping and pendulous inflorescence stem phenotype was also reported in the *AtIAA20* overexpression line, it was however interpreted as an impacted gravitropic response [49]. However, in light of our results, it is likely that these plants also lacked proper secondary fiber development to ensure the skeletal support of the inflorescence stem. The ectopic overexpression of *EgrIAA20* in *Arabidopsis* specifically uncoupled the development of the different secondary xylem cell types by inhibiting the lignification of secondary fibers but not of tracheary elements. This was only observed during secondary growth of both hypocotyls and basal parts of inflorescence stems, thereby showing that the development of fibers and tracheary elements formed in secondary xylem not only differ from primary xylem but also depend on distinct auxin-dependent regulations.

Furthermore, it would be interesting to investigate the mechanism of how non-canonical *Aux*/*IAA*, lacking the conserved degron Domain II, regulates auxin signaling. To this end, it is worth noting that another lacking Domain II non-canonical *Aux*/*IAA* member, IAA33, was shown to compete with canonical *Aux*/*IAA* member IAA5 for binding ARF10/16 to protect the latter from IAA5-mediated inhibition. Thus, IAA33 maintains root distal stem cell identity and negatively regulates auxin signaling [50]. Whether *EgrIAA20* acts through a similar mechanism deserves further attention.

Wood and fiber constitute an extremely important renewable biological resource. The efficient exploitation of this lignocellulosic biomass requires a full understanding of the wood formation process. Our results represent a timely contribution to the rapidly expanding knowledge of the wood formation regulation, especially the regulation of wood cells’ specification and maturation. The genetic engineering of wood biomass by uncoupling the development of the different cell types has been a target of recent studies, specifically to reduce lignin in fibers without altering tracheary elements. Many of these strategies use lignin monomer loss-of-function mutants that are complemented using a tracheary element specific promoter such as *ProVND6* [51] or *ProSNBE* [52,53]. We herein present another strategy to uncouple the lignification of fibers from tracheary elements by directly overexpressing *EgrIAA20.* Although our strategy enabled us to uncouple cell type specific lignification using one gain-of-function construct instead of combining loss-of-function with cell specific complementation, our strategy needs further refinement to control the level of transgene expression. Overall, our novel discovery not only opens new avenues to deepen our understanding of the mechanisms transducing auxin for cell type specific regulation in wood but also provides a proof-of-concept of a novel technological strategy to uncouple wood cell types genetically to modify lignin composition in fibers without affecting water and solutes’ conduction, thus ultimately to improve wood composition for its better end-uses.

## 4. Materials and Methods

### 4.1. Plant Material and Growth Conditions

The provenance and preparation of all *Eucalyptus* organs and tissues were as described in [54]. For phenotype characterization, *Arabidopsis thaliana* ecotype Col-0 plants (wildtype and transgenic plants) were sown on soil, vernalized for 5 days at 4 °C, then were grown in a growth chamber, under 8h-day/16h-night short days condition to promote secondary growth. The growth chamber was set to 22 °C-day/20 °C-night temperature, 70% relative humidity, 200 µmol photons m^−1^ s^−1^ light intensity (intense luminosity). The plants were watered every two days and fertilized weekly since 4-weeks-old till harvest. Seeds for in vitro culture were surface-sterilized for 1 min in 70% ethanol, 10 min in 25% bleach, rinsed five times in sterile water, and planted on MS medium containing 1.0% sucrose solidified with 1% agar.

### 4.2. Microfluidic Quantitative PCR Expression Analysis

Mature leaves, young roots, young stems, seedling, floral bud, fruits (capsules) were collected from *Eucalyptus globulus*. Shoot tips, secondary xylem, secondary phloem, and a cambium-enriched fraction were collected from 7-year-old trees of *Eucalyptus Gundal* hybrids (*Eucalyptus gunnii* × *Eucalyptus dalrympleana*, genotype 850645) grown in southwest France (Longage) by the FCBA. Cambium fractions were also collected from a 25-year-old *Gundal* hybrid (genotype 821290). Vascular tissues were sampled as previously described [55]. Briefly, a piece of bark was removed from the trunk; cambium-enriched fractions which include cambium and some xylem and phloem mother cells were obtained by gently sweeping the inner side surface of the bark and outside surface of xylem (debarked trunk). After collecting the cambium, the procedure involved scarping the inner side of bark to obtain phloem, and scarping the outer side of debarked trunk to obtain developing secondary xylem. Gene-specific primers for quantitative PCR were designed using QuantPrime [56] with default parameters (Appendix A). Transcript abundance was assessed by microfluidic qPCR using the BioMark. The reference genes were selected as described in [54].

### 4.3. Gene Cloning, Vector Construction, and Genetic Transformation

The coding sequence of *EgrIAA20 (Eucgr.K00561)*, *EgrARF5 (Eucgr.F02090)* was amplified from *Eucalyptus grandis* cDNA using high fidelity PhusionTaq (Thermo Fisher Scientific, Waltham, MA, USA), and the amplicons were inserted into the pDONR207 vector using the BP clonase II (Invitrogen, Carlsbad, CA, USA) or into the pENTR/D-TOPO vector (Invitrogen, Carlsbad, CA, USA). LR clonase II was then used to generate the different destination vectors, including pFAST-G02 [57] for over-expression of *pro35S::EgrIAA20* in *Arabidopsis thaliana*, and pGBG-BD-GTW and pGAD-AD-GTW (kindly provided by Laurent Deslandes, LIPM, Auzeville Tolosane, France) for yeast two-hybrid (Y2H). For genetic transformation, we transformed the destination vector *pFAST-EgrIAA20* into *Agrobacterium tumefaciens* strain GV3101. Then, the *A. thaliana* ecotype Col-0 was transformed using *A. tumefaciens* and the floral dip method [58]. Primary transformants were selected using a fluorescent stereomicroscope with GFP filters which is able to detect the GFP fluorescent marker present in the pFAST-G02 binary vector that is expressed specifically in transformed seeds, as described by Shimada et al. [57]. Ten independent *EgrIAA20* over-expressing T1 *Arabidopsis* lines were generated; all presented similar phenotypes showing reduced plant size as compared to the control. For detailed characterization of their phenotypes, three independent T2 lines (two strong lines and one weak line) were selected according to their T1 phenotypes and the transgene transcript abundance (determined in leaves by RT-qPCR) (Figure 4A).

### 4.4. Transient Expression in Protoplasts for Subcellular Localization and Transactivation Assay

Protoplasts for transfection were obtained from suspension-cultured tobacco (*Nicotiana tabacum*) BY-2 cells according to the method described by Yu et al. [26]. Protoplasts were transfected by a modified polyethylene glycol method as described by Abel and Theologis [59]. For nuclear localization of *EgrIAA20* and EgrIAA4, the full-length cDNAs were fused in frame at the C-terminus with GFP in the pK7FWG2.0 vector (Karimi et al. 2002) under the control of the *CaMV 35S* promoter. Transfected protoplasts were incubated for 16 h at 25 °C and examined for GFP florescence signals using a Leica TCS SP2 laser scanning confocal microscope. Images were obtained with a x40 water immersion objective. For co-transfection assays, the full-length cDNAs of the selected *Aux*/*IAA* were cloned into pGreen vector under the *CaMV 35S* promoter to create the effector constructs. The reporter constructs used a synthetic auxin-responsive promoter *DR5* followed by the *GFP* reporter gene. For co-transfection assays, aliquots of protoplasts (0.5 × 10^6^) were transformed with 10 µg of the reporter vector, containing the *DR5* synthetic promoter fused to the *GFP* reporter gene, and with either the effector vector of *EgrIAA20* under the *CaMV 35S* promoter or the empty vector as mock treatment. After 16 h incubation, *GFP* expression was quantified by flow cytometry (FACS calibur II, BD Bioscience), and the data were analyzed using Cell Quest software (BD Biosciences, San Jose, CA, USA). Transfection assays were performed in three independent replicates, and 400–1000 protoplasts were gated for each sample. GFP fluorescence corresponds to the average fluorescence intensity of the protoplasts’ population after subtraction of auto-fluorescence determined with non-transformed protoplasts. A total of 50 µM 2,4-D were used for auxin treatment.

### 4.5. Microscopy Analysis

The basal end (<1 cm) of *Arabidopsis* inflorescence stems and hypocotyls were harvested at indicated dates, and then stored in 70% ethanol. The 62-day-old plants (2 months) grown on a short day condition present fully developed inflorescence stems, and the first siliques were clearly observed. The cross sections (100 µm thick) were prepared using vibratome Leica VT1000 S (Leica, Paris, France). Lignin polymers are the characteristic components of the secondary cell wall (SCW) and are normally absent from the primary cell wall. Therefore, we used lignin deposition detection techniques to screen for the SCW phenotype. Cross sections of the inflorescence stem and hypocotyl were stained with phloroglucinol-HCl, which stains specifically the lignin polymer unit coniferaldehyde and p-coumaraldehyde in the SCW giving a violet-red color. Phloroglucinol-HCl was directly applied on the slide, and the sections were observed under a bright-field inverted microscope (DM IRBE; Leica). Images were recorded with a CCD camera (DFC300 FX; Leica) directly connected to microscope.

Maüle staining was used to distinguish vessel cells from fiber cells. Firstly, the sections were incubated in 1% sodium citrate solution of KMnO_4_ for 3 min. The stained sections were washed in abundant water for 5 min. Later, bleaching was conducted using HCl 6N for a few seconds. As soon as the sections became transparent, sections were taken out and washed in water for another 5 min. At last, 5% NaHCO_3_ was directly applied on the slices for observation under a bright-field inverted microscope (DM IRBE; Leica). Images were recorded with a CCD camera (DFC300 FX; Leica) directly connected to microscope.

### 4.6. Pyrolysis Analysis

*EgrIAA20* overexpression transgenic (*EgrIAA20_OE* lines 3.1 & 3.3) and wild type *Arabidopsis* were planted in 3 different batches in short days (5 plants for each line in one batch), and 10 days different between each batch. The hypocotyls were harvested in liquid nitrogen when the first silique was fully developed. Then, the hypocotyls were freeze-dried and ball-milled (MM400; Retsch) at 30 Hz in stainless steel jars (1.5 mL) for 2 min with one ball (diameter of 7 mm). A total of 50–70 μg (XP6, Mettler-Toledo, Switzerland) of powder were transferred to auto sampler containers (Eco-cup SF, Frontier Laboratories, Japan) for the Py-GC/MS. The sample was carried to the oven pyrolyzer by an auto sampler (PY-2020iD and AS-1020E, FrontierLabs, Nagoya, Japan) and analyzed by a GC/MS system (Agilent, 7890A/5975C, Agilent Technologies AB, Sweden). The pyrolysis oven was set to 450 °C, the interface to 340 °C, and the injector to 320 °C. The pyrolysate was separated on a capillary column with a length of 30 m, diameter of 250 mm, and film thickness of 25 mm (JandW DB-5; Agilent Technologies, Stockholm, Sweden). The gas chromatography oven temperature program started at 40 °C, followed by a temperature ramp of 32 °C/min to 100 °C, 6 °C/min to 118.75 °C, 15 °C/min to 250 °C, and 32 °C/min to 320 °C. Total run time was 19 min, and full-scan spectra were recorded in the range of a 35 to 250 mass-to-charge ratio. Data processing, including peak detection, integration, normalization, and identification, was conducted as described by Gerber et al. [60].

The relative amounts of S-, G-, and H-lignin and the carbohydrates were further expressed as the percentage of the total compound amounts. Orthogonal projections to latent structures discriminant analysis (OPLS-DA) of each individual replicate was performed using SIMCA-P+ (12.0).

### 4.7. Yeast Two-Hybrid (Y2H) Screening

The Y2H library of cambium and developing xylem from field-grown *Eucalyptus globulus* (Labill.) was constructed as described by Soler et al. [30] using the Make Your Own Mate-&-Plate Library System (Clontech, Mountain View, CA, USA). RNAs were isolated from a mixture of scraping cambium and xylem from juvenile to mature trees. The complexity of the eucalypt xylem library obtained was c. 1 × 10^6^ clones mL^−1^, with a density of *c*. 5 × 10^8^ cells.

For the library screening, the full length CDS of *EgrIAA20* was inserted into the pGBG-BD-GTW vector, transformed in Y2HGold yeast cells, and used as bait to screen the cDNA library. Y2H screening was performed using the mating protocol described for the Matchmaker Gold Yeast Two-Hybrid System (Clontech) with QDO medium (Quadruple synthetic drop-out autotrophic medium lacking tryptophan, leucine, adenine, and histidine). A total of 57 colonies were obtained and sequenced using primers designed for the pGADT7 vector (Clontech). Plasmids were isolated with the Easy Yeast Plasmid Isolation Kit (Clontech). Targeted Y2H assays to test the repeatability and the specificity of the interactions were performed using QDO/X/A medium (QDO medium supplemented with Aureoabaisin A and 5-bromo-4-chloro-3-indolyl a-Dgalactopyranoside (X-a-Gal)) and DDO medium (double synthetic drop-out auxotrophic medium lacking tryptophan and leucine), as specified by the manufacturer (Clontech, Mountain View, CA, USA).

## Figures and Tables

**Figure 1 ijms-23-05068-f001:**
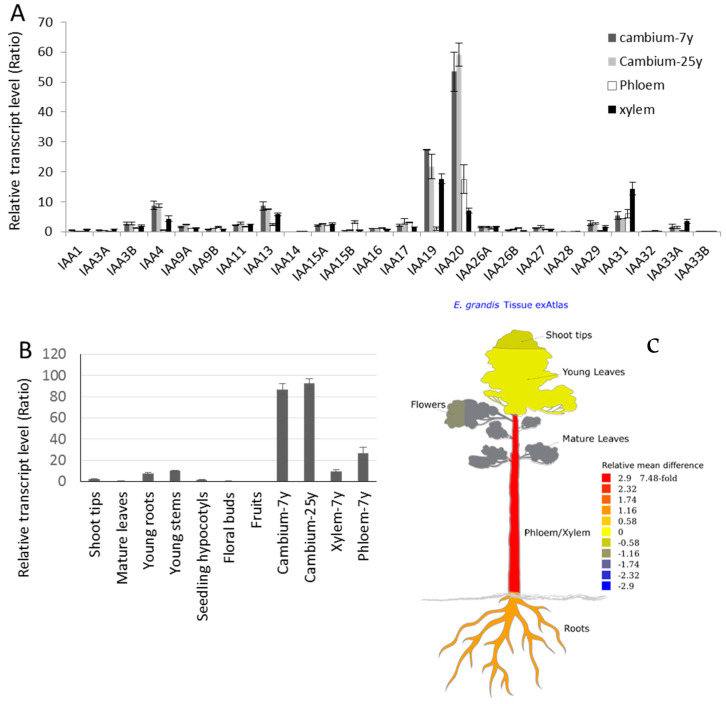
*EgrIAA20* transcript (*Eucgr.K00561*) shows preferential accumulation in the cambium of *Eucalyptus*. (**A**) Ratio of relative mRNA abundance of all *Eucalyptus Aux*/*IAA* members in vascular tissues including: juvenile vascular cambium (Cabium-7-year-old tree), mature vascular cambium (Cambium-25-year-old tree), phloem (7-year-old tree), and xylem (7-year-old tree). (**B**) Real-time PCR expression levels of *EgrIAA20* in various *Eucalyptus* tissues and organs. Each relative mRNA abundance was normalized to a control sample (in vitro *Eucalyptus* plantlets). Error bars indicate mean expression values ±SE from three independent experiments. (**C**) Pictographic view of *EgrIAA20* expression across a diverse range of RNAseq expression datasets by exImage (https://eucgenie.org/exImage, accessed on 25 April 2022).

**Figure 2 ijms-23-05068-f002:**
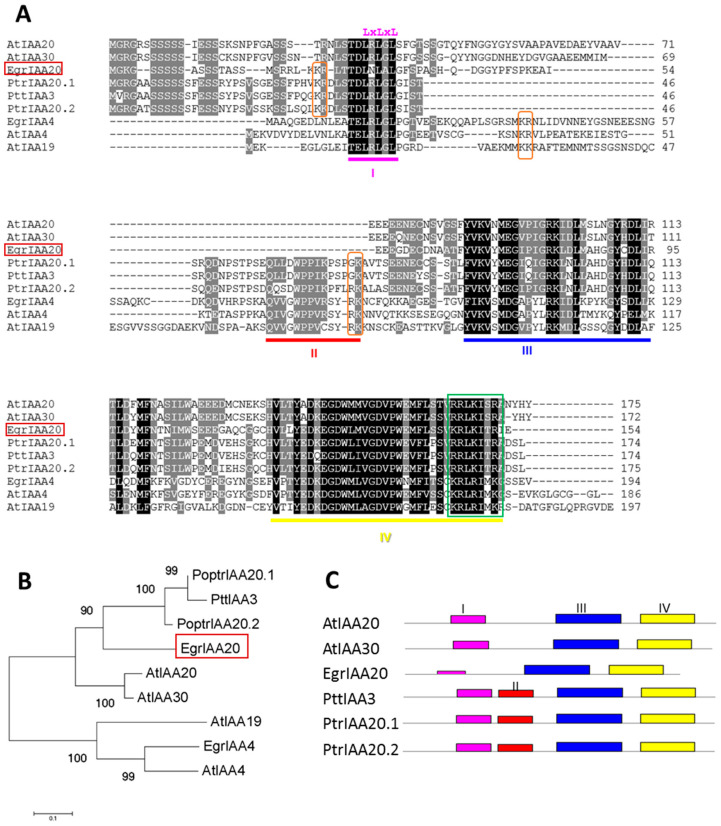
*EgrIAA20* defines a non-canonical Aux/IAA protein which lacks highly conserved Aux/IAA characteristic Domain II. (**A**) Sequence Analysis of *EgrIAA20*. Sequence comparison of *EgrIAA20*, putative orthologs from other plants, and paralog of Aux/IAA protein family. Conserved residues are shaded in black; gray shading indicates similar residues in at least 5 out of 9 of the sequences. Four conserved domains I, II, III, IV are underlined. Conserved basic residues that putatively function as NLS are indicated by orange frames (Type I) and green frame (Type II). The core repression EAR motif in Domain I is indicated by “LxLxL” on the top of the alignment. (**B**) The phylogenetic relationship of *EgrIAA20* with its related putative orthologs from *Arabidopsis* and poplar defines a distinct clade with *Arabidopsis* AtIAA20 and AtIAA30, and previously characterized wood-related aspen member PttIAA3. (**C**) Shema of proteins structures for *EgrIAA20* and its closely related putative orthologs from poplar and Arabidopsis. The domains of Aux/IAA proteins were predicted by Pfam (http://pfam.xfam.org, accessed on 23 April 2020) and are indicated by different colors; Domain II (red box) was absent for *EgrIAA20*, AtIAA20, and AtIAA30, but present in PtrIAA20.1, PtrIAA20.2, and PttIAA3.

**Figure 3 ijms-23-05068-f003:**
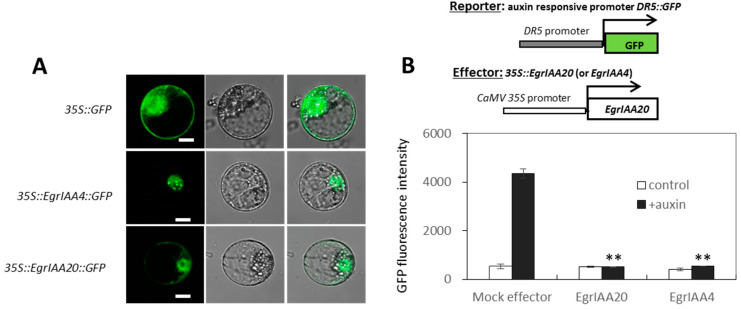
Subcellular localization and repressor activity of auxin response of *EgrIAA20* protein on a synthetic *DR5* promotor. (**A**) Subcellular localization of *EgrIAA20*-GFP fusion protein in BY-2 tobacco protoplasts. The merged images of green fluorescence (left panel) and the corresponding bright-field image (middle panels) are shown in the right panels. Canonical Aux/IAA member EgrIAA4-GFP serves as a positive control. Scale bar = 10 µm. (**B**) Transcriptional repression of auxin response by *EgrIAA20* protein on a synthetic *DR5* promotor. Effector and reporter constructs were co-expressed in tobacco protoplasts in the presence or absence of a synthetic auxin (50 µM 2,4-D). A mock effector construct (empty vector) was used as negative control, and EgrIAA4 construct was used as positive control. Three independent experiments were performed, and similar results were obtained. In each experiment, protoplast transformations were performed in independent biological triplicates. The figure indicates the data from one experiment. Error bars represent SE of mean fluorescence. Significant statistical differences (Student’s *t*-test, n > 400, *p* < 0.01) from the control are marked with **. The schemes of the reporter and effector constructs are illustrated on the top of the figure.

**Figure 4 ijms-23-05068-f004:**
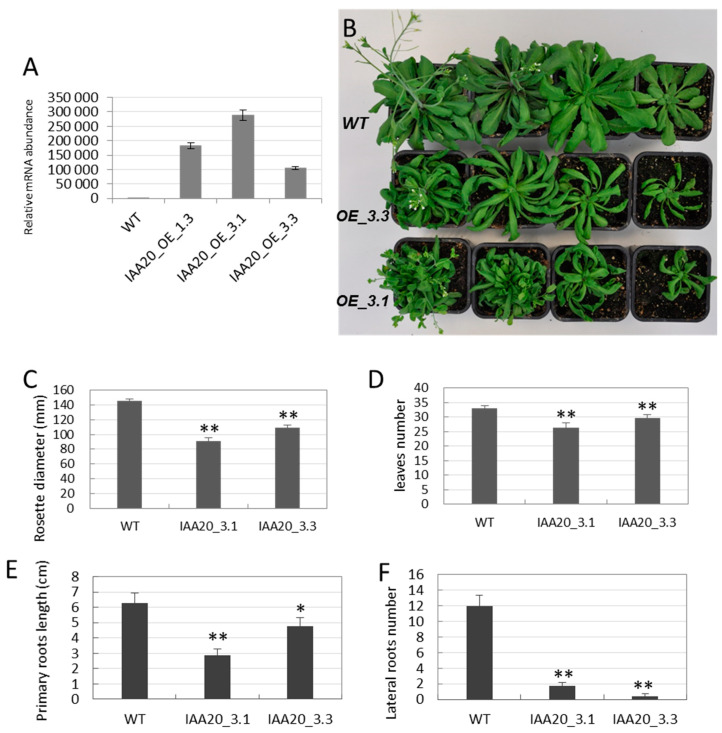
*EgrIAA20* overexpressing lines displayed reduced rosette development with helical twisting leaves and reduced root development. (**A**) *EgrIAA20* mRNA accumulation in three phenotypically representative independent transgenic lines. The *EgrIAA20* overexpressing lines (*IAA20_OE_3.1* and *IAA20_OE_3.3*) presented reduced rosette development with helices twisting and backward rolling leaf blades and reduced leaf size (**B**), significantly reduced diameter of rosette (**C**), significantly reduced leaves number (**D**), significantly reduced primary root length (**E**), and greatly reduced lateral roots number (**F**). Error bars represent standard error. Asterisks indicate values found to be significantly different from the wild-type control (Student’s *t*-test, n > 10); ** indicates *p* < 0.01, and * indicates *p* < 0.05.

**Figure 5 ijms-23-05068-f005:**
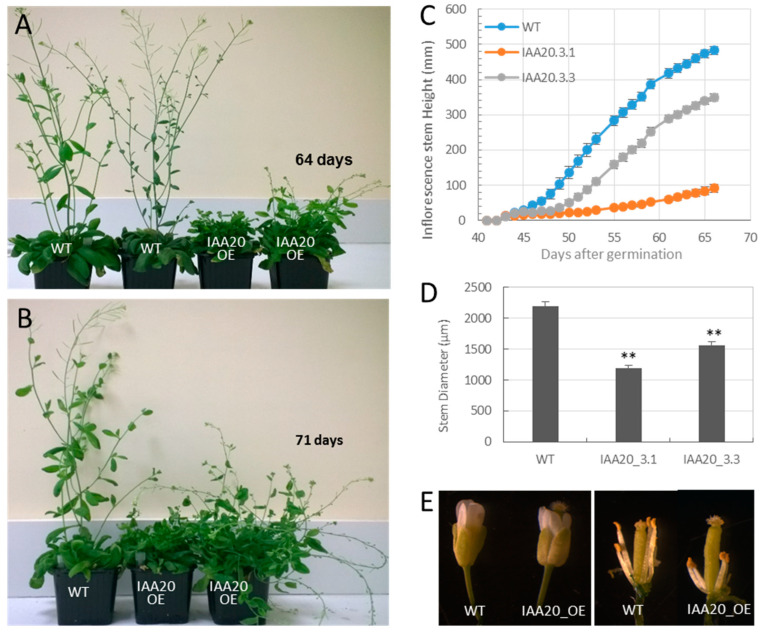
Phenotypes of *EgrIAA20* overexpressing lines. Comparison of plant architecture of aerial parts at 64 days old (**A**) and 71 days old (**B**); *EgrIAA20* overexpressing lines displayed a bushier plants architecture compared to the normal plant, with floppy inflorescence stems, in contrast to wild-type control up-right growth inflorescence stems. (**C**) Growth curve of the *Arabidopsis* primary inflorescence stem. The elongation was reduced in *EgrIAA20* overexpressing lines with significantly reduced inflorescence diameter (**D**); error bars represent standard error, n > 12; stem diameter was measured at the base (<1 cm to the rosette level) when the first silique fully developed. Asterisks indicate values found to be significantly different (student’s *t*-test, n > 12) from the wild-type control. ** indicates *p* < 0.01. (**E**) The stamens of *EgrIAA20* overexpressing lines were dramatically shorter than the wild-type control.

**Figure 6 ijms-23-05068-f006:**
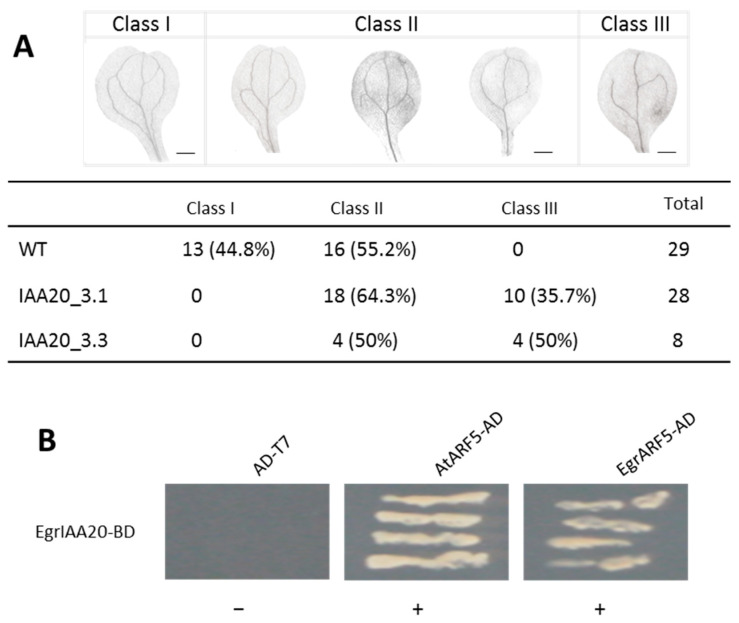
Over-expressing *EgrIAA20* in *Arabidopsis* regulated vascular patterning in cotyledons through the interaction with AtARF5. (**A**) *EgrIAA20* overexpression impaired vascular patterning in cotyledons. The vascular patterning was assessed by the number of secondary vein loops originating from the mid-vein. Class I presents four complete vascular loops; class II presents partially incomplete venation with at least two loops; class III presents entirely incompletely venation with no entire venation loop. In wild type seedlings, all the cotyledon venation patterns belonged to classes I and II, and 45% of them exhibited a more complex pattern (class I with four complete loops). In contrast, cotyledon venation patterns of all the transgenic plants belonged to classes II and III; none of them showed the complete vascular patterning with four loops (class I). Values in bracket indicate the percentage contribution of each class. The scale bars represent 0.5 mm. (**B**) *EgrIAA20* interacts with *Arabidopsis* ARF5/MP (AtARF5) and its potential ortholog in *Eucalyptus* EgrARF5 in yeast-2-hybrid assay. The *EgrIAA20* and AtARF/EgrARF5 proteins were fused with GAL4 DNA-binding domain (BD) and a GAL4 activation domain (AD), respectively. Yeast of co-transformed *EgrIAA20-BD* and *AtARF5-AD* or *EgrARF5-AD* grew on quadruple dropout medium lacking leucine, tryptophan, histidine, and adenine (THLA), and then scratched again on a TLHA plate; *AD-T7* were used as negative controls.

**Figure 7 ijms-23-05068-f007:**
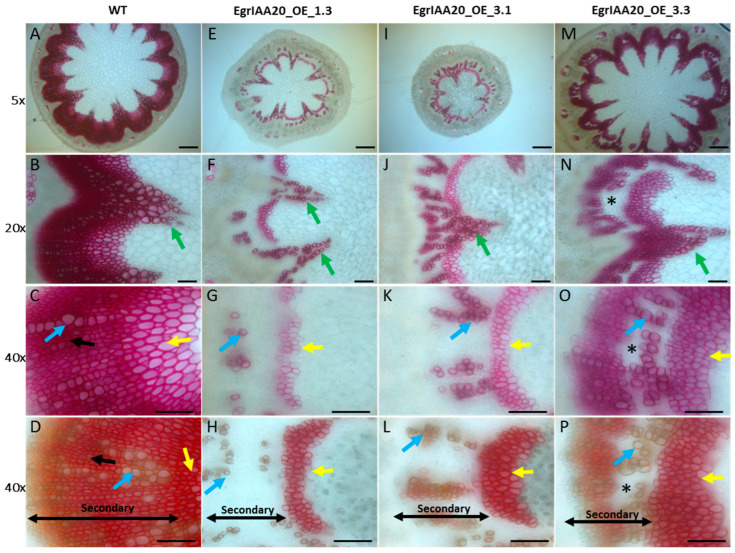
*EgrIAA20* overexpression specifically repressed secondary fibers but not primary fibers nor secondary vessels in *Arabidopsis* inflorescence stems. Left panel (**A**–**D**) shows cross section of inflorescence stems at the basal part of wild-type control, <1 cm to the rosette level, 62-day-old plants; 5×, 20×, and 40× objective image for the first raw, second and third raw, respectively, using Phloroglucinol-HCl staining, which stains the lignin polymers in the SCW of xylem cells into red-purple; the last raw was 40× objective observation image stained with Maüle method, which stains the fiber cells into bright red color due to the syringyl unit (S unit) of lignin, indicated by yellow and black arrows, and stains the vessels in to brown due to the G unit of lignin, indicated by blue arrow. The corresponding cross sections from three independent *EgrIAA20* overexpressing lines are shown in the middle and right panels: *EgrIAA20_OE_1.3* (strong line, middle left panel (**E**–**H**)), *EgrIAA20_OE_3.1* (strong line, middle right panel (**I**–**L**)), *EgrIAA20_OE_3.3* (weak line, right panel, (**M**–**P**)). Green arrows indicate primary xylem cells in fascicular bundles; blue arrows indicate secondary vessel cells (vessel II); yellow arrows indicate primary fiber cells (fiber I); black arrows indicate secondary fiber cells (Fiber II). * indicates staining gaps dispersed in the secondary growth region. The first to third raw used Phloroglucinol-HCl staining; the bottom raw used Maüle staining methods. The secondary growth regions are indicated by double-arrow lines in the images stained by Maüle. For 5× objective observation images, scale bar = 200 µm, and for 20× and 40× objective observation images, scale bar = 50 µm.

**Figure 8 ijms-23-05068-f008:**
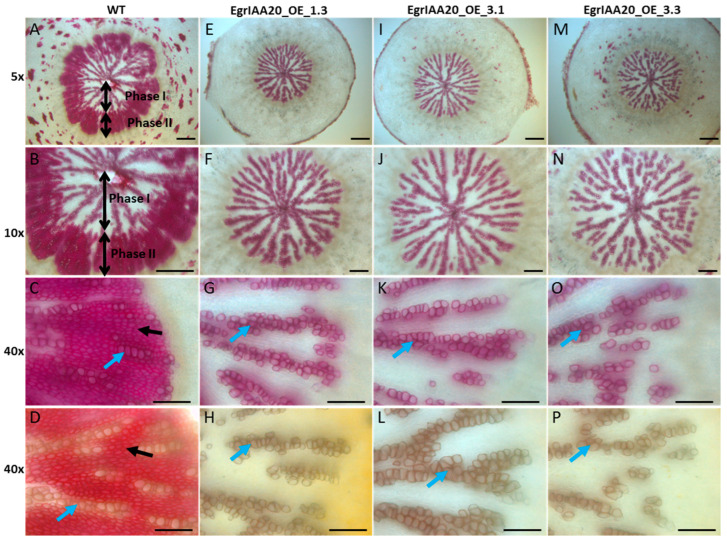
*EgrIAA20* overexpression specifically inhibits the lignification of fibers in secondary but not primary fibers nor any tracheary elements in *Arabidopsis* hypocotyl. Left panel (**A**–**D**) shows cross section of hypocotyl of wild-type control, 5×, 10×, and 40× objective image for the first raw, second and third raw, respectively, using Phloroglucinol-HCl staining, which stains the lignin polymers in the SCW of xylem cells into red-purple; the last raw was 40× objective observation image stained with Maüle method, which stains the fiber cells into bright red color, indicated by black arrow, and stains the vessels in to brown, indicated by blue arrow. The corresponding hypocotyl cross sections from three independent *EgrIAA20* overexpressing lines are shown in the middle and right panels: *EgrIAA20_OE_1.3* (strong line, middle left panel, (**E**–**H**)), *EgrIAA20_OE_3.1* (strong line, middle right panel, (**I**–**L**)), *EgrIAA20_OE_3.3* (weak line, right panel, (**M**–**P**)). Phase I growth region and phase II growth region are indicated by double-arrow lines in wild type cross section. Blue arrows indicate secondary vessel cells (vessel II); black arrows indicate secondary fiber cells (Fiber II). The first to third raw used Phloroglucinol-HCl staining, the bottom raw used Maüle staining methods. For 5× and 10× images, scale bar = 200 µm, and for 40× images, scale bar = 50 µm.

**Figure 9 ijms-23-05068-f009:**
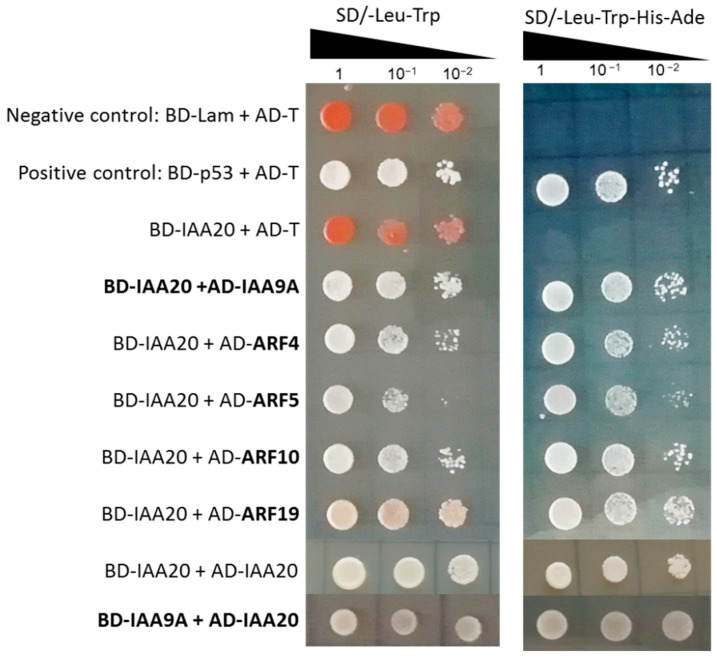
Yeast-two-hybrid analysis of protein interactions between *EgrIAA20* and xylem expressing EgrARFs and EgrIAA proteins. The *EgrIAA20* and EgrARF/EgrIAA proteins were fused with Gal4 DNA-binding domain (BD) and a GAL4 activation domain (AD), respectively. The interaction between BD-Lam and AD-T was used as negative control, while the interaction between BD-p53 and AD-T was used as positive control. Yeast cells were inoculated on selective medium in a 10-fold gradient dilution. SD/-Leu-Trp: double dropout medium lacking leucine and tryptophan; SD/-Leu-Trp-His-Ade: quadruple dropout medium lacking leucine, tryptophan, histidine, and adenine.

**Table 1 ijms-23-05068-t001:** Chemical composition of wild type and *EgrIAA20_OE* transgenic *Arabidopsis* hypocotyls by Py-GC/MS analysis (n > 12).

Genotype	Carbohydrate	Guaiacyl	Syringyl	*p*-Hydroxy-Phenol	Phenolic	Lignin	S/G	C/L
WT	77.5 ± 2.0	8.5 ± 1.2	2.9 ± 0.7	1.7 ± 0.2	1.1 ± 0.1	14.2 ± 2.0	0.34 ± 0.04	5.55 ± 0.82
*EgrIAA20* ^1^	76.9 ± 3.0	8.3 ± 0.8	2.0 ± 0.4 *	1.9 ± 0.2 *	1.0 ± 0.1 *	13.2 ± 2.4	0.25 ± 0.03 **	6.03 ± 1.4

Asterisks indicate values found to be significantly (Student’s *t*-test) different from the wild type. * *p* < 0.05, ** *p* < 0.01. ^1^: results of pooled samples from three independent batches of *EgrIAA20_OE* lines 3.1 and 3.3 (5 plants for each line each batch).

## Data Availability

Not applicable.

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
