# Peer review of "Overexpression of EgrIAA20 from Eucalyptus grandis, a Non-Canonical Aux/IAA Gene, Specifically Decouples Lignification of the Different Cell-Types in Arabidopsis Secondary Xylem"

_ijms, 2022, doi:10.3390/ijms23095068_

Round 1

Reviewer 1 Report

The manuscript of Yu et al. describes the characterization of EgrIAA20, a novel transcription factor from eucalyptus which is homologous to the Arabidopsis thaliana AtIAA20 and 30. This factor was localized mostly within the nucleus, is a strong transcriptional repressor of auxin response and interacts with EgrIAA9A which is involved in wood formation. The overexpression of EgrIAA20 in Arabidopsis triggers impaired plant development and vascular patterning and specifically the formation of secondary fibers indicating a role in cambium cell type specification and therefore in wood formation. Although some similar results were reported previously on the overexpression of the Arabidopsis IAA20 gene in Arabidopsis, this paper provides obviously novelty for improving strategies related to the targeted modification of wood properties without affect water and solutes transport in the plant. It is well written and well-illustrated. I have only few comments.

  1. IAA20 was localized mostly within the nucleus but also in the cytoplasm. What could be its function in that compartment? A sentence should be included in the discussion.
  2. Title of Figure 6 sounds not correct. There is no scale bar in Panel A. In (B) EgrIAA20 interacts
  1. Table S1: There are only 35 colonies sequenced (47 in the manuscript)

Through the text: Genes in italics: for instance, line 393

Point 4.4: gene and promoter in italics. There are 2 denominations for the 35S promoter.

Author Response

We are thankful to reviewers for having highlighted the novelty of our work for improving strategies related to the targeted modification of wood properties without affect water and solutes transport in the plant. They also helped us to better clarify our experimental results and correct some relevant grammar and typing errors. 

  1. IAA20 was localized mostly within the nucleus but also in the cytoplasm. What could be its function in that compartment? A sentence should be included in the discussion.

It was very surprising to find that EgrIAA20 was not exclusively located in the nucleus as other canonical Aux/IAA members, we supposed that it may result from the lack of type I nuclear localization signals (NLS). Weak localization signal was also found in cytoplasm, and we still have no idea about its possible function in this compartment. The no-exclusive nuclear subcellular localization was also reported for the tomato non-canonical member SlIAA32 which is also lacking the same Type I NLS, however its cytosolic function was not explored. We add a discussion sentence in the manuscript lines203-205, page 6.

  1. Title of Figure 6 sounds not correct. There is no scale bar in Panel A. In (B) EgrIAA20 interacts

We edited the title of Figure 6, added scale bars in Panel A, and corrected the grammar error in the revised manuscript, Lines 350-351 Page 11.

  1. Table S1: There are only 35 colonies sequenced (47 in the manuscript)

Thanks a lot for the careful reading. In fact we got 47 positive clones in Y2H plates initially, but we obtained results of PCR and sequencing only for 35 clones (from 47 clones). This was clarified in the revised manuscript, lines 457-459, Page 14.  

  1. Through the text: Genes in italics: for instance, line 393.

We have carefully verified throughout the text and figures: genes in italics, please see the revised manuscript as indicated by the “Track Changes”.

For example: lines 142, 196, 198, 212, 216, 228-233, 247-255, 286, 341, 364-366, 395, 424-426, 445-447, 600-606, Table 1, and Figure 3.

  1. Point 4.4: gene and promoter in italics. There are 2 denominations for the 35S promoter.

We have corrected them (gene and promoter in italics), and unified the two denominations for the promoter as “CaMV 35S” promoter throughout the revised manuscript.

Reviewer 2 Report

The authors used yeast two-hybrid assays to identify its protein partners during Eucalyptus wood formation and identified its primary interacting protein as EgrIAA9A, whose poplar ortholog PtoIAA9 is also known to participate in wood formation. They established that EgrIAA20 is a critical auxin-signaling component involved in regulating a distinct developmental stage of one type of wood cell - lignification of wood fibers. The manuscript is well structured and well discussed. However, some points should be checked and corrected before its acceptance in this journal. 

Therefore, according to my comments, I recommended the publication of the paper after minor revision.

  • Please speculate on the results. The discussion must improve.
  • Please provide in the conclusion section. The authors should add the significance of this research and its potential practical application.
  • The MS English needs to be improved. The article's English must be carefully checked for grammatical errors.

Author Response

We are thankful to reviewers for having highlighted the novelty of our work for improving strategies related to the targeted modification of wood properties without affect water and solutes transport in the plant. They also helped us to better clarify our experimental results and correct some relevant grammar and typing errors.

 “The authors used yeast two-hybrid assays to identify its protein partners during Eucalyptus wood formation and identified its primary interacting protein as EgrIAA9A, whose poplar ortholog PtoIAA9 is also known to participate in wood formation. They established that EgrIAA20 is a critical auxin-signaling component involved in regulating a distinct developmental stage of one type of wood cell - lignification of wood fibers. The manuscript is well structured and well discussed. However, some points should be checked and corrected before its acceptance in this journal.”

Therefore, according to my comments, I recommended the publication of the paper after minor revision.

  1. Please speculate on the results. The discussion must improve.

Discussion have been enriched / rephrased according to reviewers’ comments (lines 516-522, 523-527, and 562-563, page 16). We believe that the main outputs of this study are now clearly outlined in the discussion, while the minor aspects of this work are directly discussed in “results” sections, for a fluent reading of the manuscript.

  1. Please provide in the conclusion section. The authors should add the significance of this research and its potential practical application.

We added accordingly in the conclusion section of revised manuscript the significance of this research and its potential practical application. Lines 523-527, and Lines 562-563. Page 16.

  1. The MS English needs to be improved. The article's English must be carefully checked for grammatical errors.

We carefully revised throughout our manuscript from text to figures (please see the “Change track”), several native English-speaking colleagues also helped us to revise our English editing. 
